# Navigating HIV self-testing: Concerns among adolescents and young people aged 15–24 years in Uganda. An exploratory qualitative study

Richard Muhumuza[1]*, Denis Ndekezi[1], Rehema Nagawa[1], Esther Awino[1], Chiti Bwalya[2,3,4], Madalitso Mbewe[2], Joanita Nassali[1], Jackline Namara[1], Felix Rutaro[1], Musonda Simwinga[2], Virginia Bond[2,3], Janet Seeley[1,5], Andrew Sentoogo Ssemata[1,5]

1 Medical Research Council/Uganda Virus Research Institute & London School of Hygiene Tropical Medicine, Uganda Research Unit, Entebbe, Uganda, 2 Zambart, University of Zambia—Ridgeway Campus, Lusaka, Zambia, 3 Ciheb Zambia, Lusaka, Zambia, 4 Department of Behavioural and Community Health, University of Maryland, School of Public Health, College Park, Maryland, United States of America, 5 London School of Hygiene and Tropical Medicine, London, United Kingdom

* richard.muhumuza@mrcuganda.org

## Abstract

### Introduction

HIV self-testing (HIVST) has the potential to overcome barriers to conventional clinic-based HIV testing services by offering a convenient, private, and confidential way to test. This study aimed to explore the concerns about HIV self-testing among adolescents and young people (AYP) aged 15–24 years in Uganda, where HIV self-testing is still not widely available.

### Methods

This exploratory qualitative study conducted 14 audio-recorded in-depth interviews and six focus group discussions with adolescents and young people in Wakiso district, Uganda, between March 2021 to February 2022. These interviews were transcribed verbatim and analysed thematically using the socio-ecological model. All participants provided written informed consent and assent before participating in the study.

### Results

AYP viewed HIVST as a potentially helpful and acceptable testing method. However, several concerns emerged and these are presented using five themes. At the individual level, participants expressed fear of suicidal thoughts if one tested HIV positive, lack of adequate information, anticipated increased risky sexual behaviours, neglect of other HIV preventive measures, and misinterpretation of test kit results. Interpersonal concerns were centred on partner violence, parental coercion, and social rejection. At the community level, participants noted the potential for stigma, unintended

**Data availability statement:** All relevant data are within the paper and its Supporting Information files.

**Funding:** The study was supported by funding from Wellcome's Institutional Strategic Support Fund grant 204928/Z/16/Z through the London School of Hygiene and Tropical Medicine. The funders had no role in the study design and decision to publish or preparation of the manuscript.

**Competing interests:** The authors have declared that no competing interests exist.

pregnancies, and discrimination. Institutional concerns focused on the lack of referral services and inadequate counselling following HIV self-testing. At the structural level, limited accessibility for persons with disabilities was a key concern.

## Conclusion

While AYP have established that HIVST is essential, various concerns need to be addressed to improve its acceptability, up-take and utilisation. Providing more explicit information about the testing procedure, clarifying any misinterpretation of results, and providing easy access to counselling services, especially at distribution points, is crucial towards guaranteeing the effective roll-out of HIVST among young people. Special attention must also be given to marginalised populations, including people living with disabilities, to ensure fairness in their access and utilisation of services. Implementation of all this will be crucial towards maximising the potential benefits of HIVST in Uganda's HIV prevention efforts.

## Background

The HIV epidemic remains a public health concern worldwide, with 37.0 million people living with HIV, 1.3 million new HIV infections, and 630,000 AIDS-related deaths in 2024 [1]. In 2021, adolescent girls and young women (AGYW) represented 63% of all new HIV infections among young people in sub-Saharan Africa, despite being only about 10% of the population. Among adolescents, 80% of new infections were in girls aged 15–19 years, and AGYW are twice as likely to be living with HIV as young men of the same age group [2,3].

Many HIV preventive methods have been advanced that include the Abstinence, Being faithful and Condom use (ABC) strategy, Pre-Exposure Prophylaxis (PrEP), Post-Exposure Prophylaxis (PEP) and HIV Self Testing (HIVST), but despite these, HIV acquisitions continue to increase [4]. HIV self-testing is a process whereby a person who wants to know his or her HIV status collects a specimen, performs a test and interprets the test result in private [5–8].

Studies have explored HIV self-testing (HIVST) among the general and key populations across the globe, including sex workers, fisherfolk, and men who have sex with men (MSM) [9–14]. These studies found that HIVST was highly acceptable, although with low levels of knowledge about the testing process.

Although HIVST was found to be acceptable, several concerns have been reported, including increased stigma, particularly among adolescents, perceptions by communities that self-testing may lead to unintended consequences, such as social isolation, anxiety, and depression [15–18].

Additionally, HIVST has been associated with violence, particularly among vulnerable populations such as women, sex workers, and MSM in settings where gender-based power imbalances are prevalent [19,20]. In some contexts, partners may react violently if an unexpected result is disclosed, or HIVST may be coercively introduced in relationships [21,22].

There has been a high uptake and coverage of HIV self-testing (HIVST), particularly among young adults, highlighting its effectiveness in reaching underserved groups such as men, young people, and first-time testers. Despite this promising evidence, further work is needed to address gaps in access and implementation, as access to HIVST among young adults remains uneven and suboptimal [23–25]. HIV self-testing was viewed as an approach that is more youth-friendly, removed obstructive age-of- consent requirements, and when peer-led (and other forms of trusted support) can help make it easier for young people to know their HIV status [26].

Additionally, there is a limited understanding of how AYP aged 15–24 years navigate the health system to access and use HIVST effectively. Existing studies have highlighted key challenges in this age group, including failure to interpret test results and the absence of counselling following the test among adolescents [14,25], and linkage to care [27]. In the Ugandan context, additional barriers such as health facilities being perceived of as confusing, physically distant, unaffordable, and staffed by older, judgmental healthcare providers further deter AYP from seeking HIV testing services [28].

In Uganda, HIVST was initially adopted in 2018 as a testing approach for Key Populations [29,30]. HIV self-testing kits were distributed through pilot projects. The Ugandan government, with support from implementing partners, have made significant investments in HIVST services [31]. HIVST kits have been made available for sale in pharmacies, but not in public facilities [32]. However, uptake and utility by adolescents and young people in Uganda remain suboptimal, and this is likely to affect the attainment of the 95-95-95 UNAIDS goals to end HIV by 2030.

Given that HIV testing remains a critical entry point for both prevention and treatment services, understanding the concerns AYP face around HIVST is essential. This study aimed to explore concerns related to HIVST among AYP aged 15–24 years in Uganda prior to the national rollout. The insights from this study will help inform more youth-responsive HIVST strategies and contribute to improving testing uptake and outcomes among this priority population.

## Theoretical orientation

This study was guided by the Social-Ecological Model – SEM [33,34] which provides a multilevel framework for understanding how individual, interpersonal, community, institutional, and structural factors shape adolescents' and young people's concerns about HIV self-testing. The SEM model proposes that an individual's health behaviour is influenced by multi-level, interdependent factors, including individual, interpersonal, community, and broad society or system levels [35]. In the individual level concerns include suicidal tendencies, limited information, increased sexual partners, neglecting preventive measures and misinterpretation of the results. At the second level, interpersonal/network concerns relate to partner violence, rejection, forcing children by parents, and discrimination. At the community level, the concerns reveal stigma, unintended pregnancies as well as discrimination. For the institutional/health system level, concerns in line with referral services and lack of counselling are revealed. At the apex of the model are the structural concerns that include non-inclusiveness and not protecting sexually transmitted diseases (Table 2). The model is well suited for this analysis because HIV self-testing occurs within complex social and environmental contexts, and young people navigate these interconnected systems in their daily lives [36].

While applying the SEM to explore adolescents' and young people's concerns about HIV self-testing, we also used the Health Belief Model (HBM) and the Theory of Planned Behaviour (TPB) offer complementary explanatory value by illuminating the cognitive and motivational processes that operate within the individual domain. The SEM outlines how individual, interpersonal, community, institutional, and structural factors shape HIV self-testing experiences [34], the HBM helps explain how perceived susceptibility, severity, benefits, barriers, and self-efficacy influence fears such as misinterpreting results, suicidal ideation, or neglecting preventive measures [37,38]. Similarly, the TPB provided insights into how attitudes, subjective norms, and perceived behavioural control influenced behavioural intentions in contexts marked by peer dynamics, partner influence, and parental authority [39]. Integrating HBM and TPB within the SEM framework strengthened the analysis by linking multilevel ecological influences with the psychological determinants that drive adolescents' and young people's responses to HIV self-testing in Wakiso district.

## Methods

### Description of the study setting and study design

This exploratory qualitative study formed part of a broader mixed-methods project investigating the knowledge, acceptability, and social implications of a peer-to-peer HIVST distribution model among adolescents and young people aged 15–24 years in Uganda. The study was conducted between March 2021 and February 2022 in three fishing landing sites: Kigungu, Gerenge, and Nakiwogo, in Wakiso District. These communities in Uganda and elsewhere are characterised by a highly mobile and predominantly young and vibrant population engaged in fishing and related hospitality-sector work (e.g., bars, lodges, and restaurants) [40,41]. Adolescents and young people in fishing communities experience unique vulnerabilities, including high mobility, economic instability, transactional sex, and elevated HIV-acquisition risk [42–45]. These contextual factors make them an important population for exploring the potential benefits, challenges, and concerns surrounding HIVST as an HIV preventive measure.

### Study population

The primary population for this study was drawn from the broader HISTAZU study, which assessed the knowledge, acceptability, and social implications of a peer-to-peer HIV self-testing kit distribution model among adolescents aged 15–24 years in Zambia and Uganda [46]. For the current analysis, we focused on male and female adolescents and young people aged 15–24 years residing in communities selected from the three study sites in Wakiso District in Uganda.

### Participant selection and recruitment

Participants were purposively recruited through the village information meetings and peer-mobilisation activities which was an efficient way to reach the target population in the fishing community. The Community Health Extension Workers (CHEWs), formerly Village Health Teams (VHTs) supported only with the community mobilisation by informing the adolescents and young people about the scheduled community meetings. We worked with the CHEWs as they play a central role in community-based health services particularly in providing sensitive reproductive health and HIV care in Uganda [47,48]. During these meetings, the research team provided information about the study, after which individuals who were interested voluntarily approached the team to express their willingness to participate, where they received detailed information about the study, including their right to decline participation without penalty. Consent and assent processes were conducted in private to ensure voluntariness and privacy. To protect confidentiality and minimise any perception of coercion or influence, CHEWs did not attend the information meetings and were not involved in the consent process, participant registration, or any subsequent study procedures. All the enrolments were conducted privately by the research team, ensuring that participation was entirely voluntary and that no identifying information was shared with CHEWs or community leaders.

### Data collection

Data for this study were collected through Focus Group Discussions (FGDs, n = 6) where 60 AYP participated. Each FGD was conducted by two researchers: one moderating the discussion with the other observing and taking notes of the proceedings. The FGDs were a useful method for collecting general community perspectives and shared experiences related to HIVST.

To yield additional detailed insights, key themes explored in the FGDs were further explored through the in-depth interviews IDIs (n = 14) with the goal of obtaining detailed narratives of AYP's views on HIVST as some AYP perceived the topic of HIV and testing as sensitive.

These FGDs and IDIs explored AYP perceptions of, as well as preference for, HIVST distribution models, points of access, and type of test kit (blood-based or oral). They further explored perceptions of and experience with barriers and facilitators of access to and use of HIVST among AYP in Uganda.

Utilising both FGDs and IDIs enabled the researchers to navigate a wide variety of different views about a particular issue and provided the opportunity for the researchers to observe how individuals collectively made sense of HIVST and the meanings they attached [49]. The interview guide was translated from English to Luganda (a local language widely spoken in the study area) by a language professional and back translated to ensure conceptual equivalence and cultural sensitivity. The translation of the guide was to ensure that the questions were asked uniformly during the data collection process. The interview guides were pretested with AYP from a different fishing community to check for accuracy before the actual data collection. It is important to note that using audio recorder, thorough probing, and interviewing up to data saturation were done to ensure dependability of the data.

## Data management and analysis

All the IDIs and FGDs were audio-recorded, transcribed verbatim and translated into English (for those in Luganda). The transcripts, alongside the audio recordings and notes taken during the data collection, were reviewed to ensure consistency and that meaning was not lost during the transcription and translation. The data were uploaded onto the server for secure storage.

We conducted the analysis following Braun and Clarke's six phases of thematic analysis [50,51]. First, five members of the research team (RM, DN, EA, RN and AS) familiarised themselves with the data through repeated reading of transcripts and field notes to understand the context of adolescents' and young people's experiences with HIVST. Initial codes were then generated inductively from the data. These codes were compared and discussed in regular analytical meetings, during which discrepancies were resolved and related codes were grouped into broader categories. Emerging themes were subsequently reviewed for coherence within and across coded excerpts, then defined and named to capture their central meanings. The final analytical narrative was developed by integrating the refined themes within the study context.

To ensure credibility, multiple coders (RM, DN, RN, EA, JN, FR and AS) in two groups participated throughout the analytical process, and debriefing meetings were held with the wider research team to compare interpretations and validate coding decisions. An audit trail including coding matrices, analytical notes, and documentation of decision-making was maintained to support transparency. A final coding framework was applied manually using Microsoft Excel. Although coding was primarily inductive, the SEM framework functioned as a sensitising framework following an abductive approach. The SEM did not determine codes; instead, it was used to guide attention to influences operating at individual, interpersonal, community, and structural levels when reviewing and organising themes, without constraining emergent insights.

## Positionality and reflexivity statement

This study was conducted by a multidisciplinary team of experienced Ugandan, Zambian, and British researchers. The Ugandan team, who collected the data, brought insider knowledge of the local culture, language, and community dynamics, which facilitated rapport-building, trust, and access to adolescents and young people in the study sites. The Zambian and British team members, who were not involved in data collection, provided outsider perspectives that enhanced reflexivity during analysis and helped identify assumptions or biases that may have been embedded in interpretation. Reflexive practices, including weekly peer debriefings and member checking, were employed to critically examine how researchers' positions, prior experiences, and beliefs influenced questioning strategies, probing, and interpretation of participants' narratives. This combination of insider and outsider perspectives strengthened the credibility of the findings and contributed to a nuanced understanding of adolescents' and young people's experiences with HIVST.

## Ethical considerations

This study received approval from the Uganda Virus Research Institute Research and Ethics Committee (GC/127/20/05/767), the Uganda National Council for Science and Technology (SS446ES), and the London School of

Hygiene and Tropical Medicine (ref 22588). Participants above 18 years provided written informed consent, while written parental consent and assent were obtained from those below 18 years before participating in any study procedures.

## Findings

### Socio-demographic characteristics

A total of 82 participants took part in both the in-depth interviews and group discussions, primarily adolescents and young adults aged 15–24, with the largest proportion falling in the 19–22 age group. The demographics are presented in Table 1.

During the analysis, we identified key themes describing concerns among adolescents and young people. These were categorised into five levels (individual, interpersonal, community, institutional and structural) based on the socio ecological model constructs as represented in Table 2.

### Individual-level concerns with HIV self-testing

The AYP concerns about self-testing at the individual level included anticipated social harm because of self-testing, failure to link with care and treatment, perceived increased sexual behaviour among AYP, and misleading results due to failure to interpret the test.

**Table 1. Participant demographic characteristics.**

| Category | Sub-category | Frequency | Percentage (%) |
|---|---|---|---|
| Age Group | 15–18 | 26 | 31.70% |
| | 19–22 | 31 | 37.80% |
| | 23–24 | 25 | 30.50% |
| Sex | Male | 38 | 46.30% |
| | Female | 44 | 53.70% |
| Education Level | Primary | 14 | 17.10% |
| | Secondary | 60 | 73.20% |
| | University | 8 | 9.80% |
| Occupation | Student | 49 | 59.80% |
| | Self-employed | 5 | 6.10% |
| | Casual worker | 3 | 3.70% |
| | Hairdresser | 2 | 2.40% |
| | Potter | 1 | 1.20% |
| | Builder | 1 | 1.20% |
| | Artist | 3 | 3.70% |
| | Fish monger/Fisher man | 18 | 22.00% |
| Marital Status | Single | 80 | 97.60% |
| | Married | 2 | 2.40% |
| Religion | Muslim | 30 | 36.60% |
| | Catholic | 25 | 30.50% |
| | Born again | 17 | 20.70% |
| | Protestant | 10 | 12.20% |
| Tribe | Muganda | 30 | 36.60% |
| | Musoga | 10 | 12.20% |
| | Mutooro | 9 | 11.00% |
| | Munyankore | 18 | 22.00% |
| | Lugubara | 5 | 6.10% |
| | Other | 10 | 12.20% |

**Table 2. Summary of the results.**

| SEM Levels | Theme | Sub themes | Quotes |
|---|---|---|---|
| Individual level | Social Harm | Suicidal tendencies<br>Perceived increased sexual behaviour<br>Failure to interpret results | "That is also not good. It has its disadvantages. You can test yourself and turn positive and, in the process, commit suicide. So, you need to carry out the test with someone who is an expert and can counsel you in case of anything." |
| Interpersonal Level | Stigma | Fear of violence<br>Relationship breakdown<br>Coercion to test | "Parents will start bringing them home and force us to test for HIV" |
| Institutional | Lack of Pre and Post test counselling | Lack of counselling services<br>Lack of referral for confirmatory tests<br>Referral for treatment | "Yes, with self-testing, there is no counselling… adolescents do not know many things. We still need counselling to be informed of the current information concerning HIV/AIDS so that we don't live the same lifestyles. In this current era, we need to be ready for everything." |
| Structural level | Complexity of use | Suitability for People with disabilities | "A blind person can't read the result because even if he tests himself, how is he going to interpret the results? Yes, it is not good for blind people. |

Participants expressed concerns about the emotional and social consequences of HIVST, particularly in the absence of counselling. Many anticipated intense psychological distress, including suicidal thoughts, following a positive test result, particularly among vulnerable youth as one noted:

*"In case results are out and one has not received enough counselling, some people develop fear and lose hope, and in the end, they end up thinking of committing suicide because they see no meaning in life. They must receive enough counselling to accept the situation." IDI Male 19-22 years*

This was attributed to the absence of pre- and post-testing counselling with HIV-self testing exacerbating sentiments of despair and psychological disorientation.

*"That is also not good. It has its disadvantages. You can test yourself and turn positive and, in the process, commit suicide. So, you need to carry out the test with someone who is an expert and can counsel you in case of anything." IDI Male 15–18 years*

This concern reflects internalized stigma and social pressures, where a positive HIV result threatens both adolescents' sense of self-worth and their perceived social standing. The fear of social exclusion and judgement illustrates how psychological responses to HIVST are embedded within broader social norms about HIV and illness.

**Changes in sexual lifestyle.** The participants also described anticipated changes in sexual behaviour where HIV self-testing would increase risky sexual behaviour among their peers, particularly unprotected intercourse. This worry stemmed from the assumption that AYP who test HIV-negative would engage in more casual and unprotected sexual encounters, potentially leading to increased risky sexual behaviour.

*"Sexual activity and risk-taking among adolescents will increase. As you said, once adolescents know their sexual partner's HIV status is negative, they will engage in unprotected sexual intercourse." IDI Male 15–18 years*

Such concerns indicate the interplay between perceived risk, peer norms, and moral expectations where the adolescents anticipate that HIVST may alter behaviours based on assumptions about HIV status.

Additionally, some participants emphasised the importance of integrating risk reduction messaging into HIV self-testing programs. Similarly, providing accessible sexual health services and monitoring the impact of self-testing on adolescent sexual behaviour would be critical in preventing such unintended consequences, as one participant narrated:

*"I think they should teach us more about HIV self-testing and prevention when they give us these self-testing kits. If I test negative, I don't want to get complacent and forget to use condoms. We need to know how to protect ourselves and our partners." GD Female 19-24 years.*

Other AYP considered HIVST as a source of empowerment, creating a sense of positive control over one's sexual life-style and health, especially in times when one feels at heightened risk of HIV.

*"Most cases when adolescents know they are HIV negative; sexual behaviour also reduces because they want to remain HIV negative. They do not over engage in sexual intercourse because they know that when they do engage in sexual intercourse they are at risk of contracting HIV. In most case when adolescent sexual partners know each other's status, it increases faithfulness and trust in that couple." IDI Male 23–24 years*

Some of the participants further expressed that HIVST will provide an opportunity for young people to adopt more responsible sexual behaviour responses and preventative lifestyles, viewing the HIVST model as an additional protector.

**Inability to interpret test results.** Misinterpretation of test results emerged as another critical concern. Participants reported that difficulties in understanding HIVST outcomes could lead to false reassurance or denial, potentially delaying care. For example, some of the participants noted that incorrect use or interpretation of HIVST could yield false negative results, leading individuals to forego linkage to care and antiretroviral therapy.

*"HIVST can mislead some people, especially when used incorrectly and produce wrong results that they are HIV negative, yet they are HIV positive, thus end up not linking to care to get medication." GD Males 15–17 years*

*"For someone who doesn't understand how to use the kit, when it reaches T and gets stuck, he or she might not understand whether it is positive or negative." GD Females 15–17 years*

Furthermore, some participants, particularly those with limited literacy, expressed scepticism about HIVST due to difficulties in understanding test results, opting instead for standard clinic based rapid testing conducted by healthcare professionals.

*"Some adolescents will prefer to use the standard test because they do not know how to read or understand the results after testing and thus do not trust the self-testing kit." IDI Male 23–24 years*

Beyond misinterpretation of results, participants raised concerns about the denial of the positive test. This was linked to the premise that some AYP would not be able to believe in the test results from the self-testing kits. For many AYP, blood-based HIV testing is what was trusted to provide accurate test results, as HIV was commonly assessed through blood and not saliva. additionally, they found it unusual that an oral saliva-based swab test could be used to test for a virus found in the blood.

*"Some of us find it a challenge to accept the truth. You might turn positive and think that the kit was faulty. Other young people will ask you if you have ever seen HIV/AIDS tested through saliva or if HIV/AIDS is transmitted through saliva because the kit does not use blood during testing." IDI Male 15–18 years*

**Neglect of other HIV preventive methods.** Participants mentioned HIVST would offer relief from the discomfort of using condoms to protect themselves against HIV after testing negative with the kit. This meant that they were more likely to abandon other protective measures. AYP also noted that when one tests with his partner using the kits and both get a negative result, they find no reason to engage in sexual intercourse with a condom.

*"The kits may decrease use of condoms because people who test negative will have unprotected sex thinking that they are HIV negative and end up acquiring other infections." GD Males 18–24 years*

Some participants recognised that HIVST kits, unlike condoms, would not prevent Sexually Transmitted Infections (STIs). This pointed to the fact that condoms were used to prevent other related risks such as STIs. Those who tested negative would be influenced to engage in sex without any other protection, thereby increasing their risk of contracting STIs.

*"What I can tell you is that there is going to be an increase in infection rates related to STDs. Apart from HIV, these other STIs are going to increase because most adolescents' concerns are about HIV it is not all about HIV. Once they test Negative, they are going to have unprotected sex, and this will put them at high risk of contracting these STIs." GD Females 18–24 years*

Similarly, unprotected sex would lead to increased cases of unintended pregnancies among adolescents and young people. Some participants argued that after testing negative, they would find no need to use condoms, leading to unintended pregnancies, as the following excerpts reveal:

*"After testing with your partner and finding out that you are HIV negative, there is no need to wear a condom. But the problem is that it is going to cause Pregnancies." GD Males 18–24 years*

*There will be an increase in unwanted teenage pregnancy. After self-testing and knowing that, they are both HIV negative. They will not use condoms; what will happen next will be an unplanned pregnancy." IDI Male 15–18 years*

The unplanned pregnancies resulting from unprotected sex would lead to increased abortion rates, as many young girls are unprepared for pregnancy and family responsibilities.

*"Since they are going to have unprotected sex the rate of abortion is going to increase because most of these girls don't want pregnancy and are not ready for family issues" GD Females 18–24 years.*

**Interpersonal level concerns.** At the interpersonal level, the AYP participants highlighted concerns about coercion, relationship conflict, and violence. Some feared negative reactions from partners upon disclosure of a positive result. HIVST was therefore viewed as a bad idea for couples to test themselves without any supervision and counselling.

*"Self-testing will cause domestic violence in homes. For example, among married couples, self-test for HIV/AIDS with your partner, and you are the only two in the house. In addition, the result comes out, one is positive, and another is negative. At that moment, a man can physically abuse the woman and even separate without getting proper counselling." IDI Female 15–18 years*

*"Self-testing is going to separate couples. From what I see is your partner will bring the test kit and want you to test together. Yet you are HIV positive, but you lied you are negative. Then you will regret why they invented self-testing kits. Because of this, relationships are going to break." IDI Female 15–18 years*

The sudden revelation of your HIV status may frequently result in relationship breakdowns underscoring the need for comprehensive counselling and support services while using HIVST to mitigate conflict risks. These concerns reveal underlying gender norms, power imbalances, and the social consequences of nondisclosure, emphasizing the complex negotiation of autonomy, privacy, and safety in intimate relationships.

Additionally, some of the participants expressed concerns that HIVST kits would facilitate coercive testing by parents, compromising their autonomy and privacy, and undermining their ability to make independent decisions about their health.

*"Parents will start bringing them home and force us to test for HIV" GD Female 15–18 years.*

While some worried about parents forcibly administering home tests, others anxiously contemplated explaining positive results, anticipating parental judgement and overreactions.

*"In case I turn positive, what will I explain to my parents? They might think about many things," IDI female 15–18 years.*

This dual concern highlights the delicate dynamics of authority and control within households and family relationships, highlighting how family structures and societal expectations can compromise adolescents' agency in health decision-making.

## Institutional-level concerns

**Absence of counselling and linkage to care.** The participants in our study expressed concerns about the absence of structured counselling, support, follow-up and linkage to care after HIVST. This was discussed in three aspects: absence of counselling, absence of referral services for confirmatory testing and failure to link to care and further treatment, as well as fear of collecting medication from the service providers.

The majority of the participants highlighted HIVST as an important HIV testing strategy for AYP; however, the lack of counselling services if an individual tested positive creates fears and concerns in the uptake of the HIVST model.

*"HIVST would be good only that it lacks counselling which is very important because once you discover that you are positive, you can end up in deep thoughts [anxious] unlike when you access counselling where you can be helped and guided on how to start medication so it will be hard to use the HIVST method" FGD Male 18–24 years.*

Participants believed that a person must be prepared to receive the outcome from the results of the HIVST kit. The participants felt that they would have nobody to reach out to, which was seen as dangerous, as somebody would collapse, engage in risky sexual behaviours and get depressed because of a positive HIV self-test.

*"Some people may struggle to cope after self-testing and may turn to alcohol. For such individuals, self-testing may not be appropriate. At the hospital, when someone is unwell, they receive counselling and emotional support to ensure they begin ART. But with self-testing, there is no one to offer comfort or guidance you are left to cry alone until it becomes too much..." IDI Male 23–24 years.*

*"Yes, with self-testing, there is no counselling… adolescents do not know many things. We still need counselling to be informed of the current information concerning HIV/AIDS so that we don't live the same lifestyles. In this current era, we need to be ready for everything." IDI male 15–18 years*

Referral and linkage for confirmatory testing was interpreted as a structural barrier that heightens psychosocial vulnerability and may exacerbate risky behaviours. Some of the participants revealed a fear of going to the facility for a confirmatory test. It would therefore take a long process of convincing somebody to go for a confirmatory test.

*"Some partners fear going to the hospital and because of that, I can try my level best to convince her to test. I can also tell her that in case she didn't understand, she can go to the health worker for more clarity, and we can try it out. So, in the process of testing, I also tell her that there is something I didn't understand and convince her that we go for a confirmatory test at the health facility." IDI Male 19–22 years*

*"That is what I was telling you if someone tests HIV positive, he or she will fear to come to the health facility to confirm her or his results. "Why would I go to the hospital, yet I already know my results". Should I go to the hospital, and they give me all that medication which is supposed to be taken every day!!" IDI Male 19–22 years*

These narratives suggest that institutional deficiencies intersect with individual apprehensions and interpersonal dynamics, shaping overall perceptions of HIVST safety and acceptability. Integrating self-testing with comprehensive support, training healthcare providers in adolescent-friendly care, and developing targeted interventions can help ensure timely linkage to antiretroviral therapy and improve health outcomes among adolescents living with HIV.

### Structural level concerns

**Complexities of the use of HIVST.** The HIV self-testing kits were viewed as posing significant accessibility challenges for visually impaired individuals, particularly the blind, who face difficulties in reading and interpreting results and lack autonomy in testing. AYP noted that these current kits require visual interpretation, excluding those who cannot see, and reliance on others for result interpretation compromises confidentiality and increases stigma.

*"This model is going to be hard for the blind because she can collect the samples very well but cannot interpret the results." GD Females 15–18 years*

*"A blind person can't read the result because even if he tests himself, how is he going to interpret the results? Yes, it is not good for blind people. IDI Female 19–22 years*

*It is true. The self-test kit is not user-friendly for the blind because there is no way a blind will know whether he is positive or negative." IDI Female 15–18 years*

These concerns point to inequities in health technology design and emphasize the need for inclusive HIV prevention tools. Beyond technical usability, these challenges also carry social implications, as reliance on others to interpret results compromises confidentiality and reinforces stigma. These findings not only highlight the psychological risks associated with receiving a positive result through HIV self-testing but also reveal critical gaps in the accessibility of self-testing kits for certain populations.

## Discussion

HIVST is a new approach to HIV testing and was rolled out in Uganda in 2018 among adults [52] but at the time of our study, HIVST had not been rolled out to the general population. We, therefore, set out to understand adolescents and young people's concerns about HIVST once rolled to the entire population. Despite UNAIDS recommendation for self-testing across all age groups, over 70% of adolescents and young people were unaware of HIVST [3]. Both older and younger adolescents had similar concerns and what came out most clearly was the feeling of self-harm if one tested positive. This means that the participants in HIVST were seen to be more likely to be depressed and consequently commit suicide or even harm the person they anticipate having infected them with HIV. This was also linked to non-disclosure and consequently failure to link to care which contravenes the 95-95-95 as stipulated by UNAIDS and World Health Organisation. Our findings also highlight the role of developmental transitions in shaping adolescents' and young people's perceptions of HIVST, consistent with a chronosystem perspective within the SEM. Younger adolescents often expressed heightened anxiety about parental discovery, anticipated social harm, and potential psychological distress, reflecting the influence of family and social norms on their perceived risk. In contrast, older youth navigating transitions into work, sexual partnerships, or early parenthood interpreted HIVST as an opportunity for empowerment and greater control over sexual health decisions. These age-related differences illustrate how evolving social roles, responsibilities, and exposure to

sexual networks interact with perceived stigma, shame, and social pressures to influence anticipated outcomes of HIVST. This abductive approach allowed emergent insights to refine our understanding of HIVST concerns without forcing data into predetermined categories, underscoring the need for age- and context-sensitive counselling, support, and linkage mechanisms.

The anticipation of social harm among adolescents and young people in our study was consistent with results from other populations [9,53] who found that people feared that there was a likelihood of a person committing suicide upon learning of positive results and lacking counselling.

Our findings support a review by Brown, Djimeu, and Cameron [54] that highlighted concerns about potential unintended harm, including psychological harm when testing and counselling are decoupled, social harm from the potential unethical use of HIV self-test kits or a nonreactive (negative) HIV self-test resulting in justification for unprotected sex, and medical harm from greater potential for inaccurate results. It is imperative to note however that from the review articles, the potential concern of suicide with HIVST was mentioned, but no evidence of suicide after an HIVST was reported [54]. A systematic review by [55] revealed that large-scale trials and implementation of HIVST have yet to identify any reports of suicide or self-harm following HIVST. Other issues reported by studies conducted in Kenya, Zambia and Uganda reflected that participants there were concerned with the fact that HIVST would breed gender-based violence as well as mistrust and suspicion among couples [19,56,57].

The absence of counselling, referral and linkage to care reported in our study where HIVST has not been rolled out to the general population continues to highlight the need to examine and evaluate the place of HIV pre-test and post-test counselling. From the findings, the concerns about anticipated social harms and a perceived increase in risky sexual behaviour are more definite when testing occurs in the absence of counselling [54].

Our findings affirm that additional research is needed to develop innovative and effective mechanisms to inform and support those using HIVST to seek follow-up confirmatory testing services, promote linkage to counselling and HIV reporting, care and partner notification [55,58].

With the great potential to reach individuals who have never tested and otherwise may not report to a health facility for testing, HIVST is key to many prevention interventions including behaviour change communications to reduce risky behaviour [54]. However, linkage to care was a contested issue where the test result came out positive. As other scholars have found, there was low linkage to care between men who have sex with men and transgender women [59]. Some of the reasons for low linkage to care were attributed to the failure to evaluate the number of people who self-test for HIV as well as the lack of counselling [13,60,61]. Another study also revealed that HIVST removes the opportunity for clinical testing to evaluate linkage to care [13].

Our study also showed that implementation of HIVST will raise concerns related to increased risky sexual behaviours. Some participants reported an increased number of multiple and concurrent partners in obtaining non-reactive results. AYP explained that unknown HIV status has been hindering them from engaging in sexual behaviours and now that they could carry out HIVST this will be an opportunity to have multiple sexual partners. Similar results were reported by Tonen-Wolyec et al. [25] where individuals who had one HIV test had fewer sexual partners than individuals who had tested five or more times. It is important to have a non-reactive screening test before engaging in sexual activities, however, AYP must be encouraged to engage themselves in other HIV testing services to reduce their risk of acquiring HIV. The results also indicated that on testing negative, AYP are more likely not to use HIV prevention methods like condoms with their sexual partners an aspect they highlighted to increase not only the number of unplanned pregnancies but also sexually transmitted infections. Another study conducted among female sex workers also revealed similar findings where some of the sex workers stopped using condoms with their boyfriends when they obtained non-reactive self-test results [62].

In addition, participants in our study believed that the introduction of self-testing services would lead to a deliberate spread of HIV. The view is that self-testing would create room for malicious AYP to spread HIV to their sexual partners

after testing positive with the kit. The increased risks of HIV and STI were linked to the absence of HIV counselling services in the model. Similar results were reported in an exploratory study in Tanzania [63]. Therefore, when rolling out HIVST among AYP, implementers must integrate it with other HIV services like HIV counselling, STI screening and treatment to help adolescents live safer and healthier lives.

While participants in this study expressed concerns about difficulties in interpreting or potentially misinterpreting the test results especially among AYP with lower literacy levels or limited knowledge of the kit recent evidence from a systematic review on HIVST suggests that laypersons can accurately perform the test with little or no supervision from healthcare providers [64]. However, the authors of the review cautioned that accuracy in reading the test results must still be closely monitored to prevent errors, particularly among those with limited knowledge or literacy [64]. In addition, the study revealed that an AYP using a HIVST is more likely to deny positive results on screening with the kit, this was linked to the notion that the kit may be spoilt or be reactive to any virus but not HIV.

The concern that HIVST kits were perceived as discriminatory underscores the need for more inclusive and accessible testing options that accommodate diverse populations, especially for those with visual impairments. This highlights a significant gap in the current design and distribution of the kits, suggesting that they may not fully meet the needs of all users, particularly those with disabilities or other specific conditions [65]. Addressing this issue could inform future policy changes and influence the development of more universally accessible HIVST kits. Notably, no other studies in the literature have reported this concern, making this an important contribution to the discourse on HIVST. Additionally, the study revealed that many AYP expressed doubts about the accuracy of oral HIVST kits, as they believed that HIV could only be detected through blood samples. This misconception points to a need for improved education and clear communication about how different HIV testing methods work [66,67]. These findings have significant implications for the design of HIVST interventions, as addressing both the perceived inclusivity and accuracy of the kits could enhance their acceptability and uptake among young people. Our results agreed with other studies from Uganda that revealed concern about the accuracy of the results [68,69].

This study has several strengths. First, the use of two qualitative methods enabled us to capture rich, nuanced perspectives from adolescents and young people and allowed for triangulation, thereby enhancing the rigour and credibility of our findings. Second, the study was conducted prior to the national rollout of HIV self-testing, providing timely insights that can inform youth-focused HIVST implementation strategies. Third, the inclusion of participants across diverse communities within the study sites strengthened the contextual relevance of the findings.

However, several limitations should also be noted. The selective sample drawn from an ongoing randomised controlled trial may not fully represent the wider population of adolescents and young people in these settings. In addition, the relatively small sample size, while appropriate for qualitative inquiry, may limit the transferability of the results. Finally, the reliance on self-reported information may have introduced recall error or social desirability bias, particularly given the sensitivity of discussing HIV self-testing.

## Conclusion

This study highlights that while HIVST is acceptable and holds substantial potential for expanding testing among AYP, several concerns remain regarding social harms, misinterpretation of results, and limited linkage to care. Addressing these concerns is essential for maximising the benefits of HIVST among AYP in Uganda. To ensure effective implementation, HIVST roll-out efforts should integrate clear educational materials, strengthened support for counselling and confirmatory testing, and mechanisms to promote timely linkage to prevention and treatment services. We recommend that future implementation efforts should prioritise ongoing monitoring of social and behavioural impacts that will also be critical to safeguarding users and maintaining trust in the intervention thereby optimise the long-term success of this strategy. Overall, HIVST represents a promising strategy as a key tool in HIV prevention and care for improving HIV testing uptake among AYP, provided that implementation is responsive to the concerns identified in this study.

## Supporting information

**S1 File. Interview guides.**
(ZIP)

## Acknowledgments

We acknowledge our study participants as well as the field mobilisers who made this study possible.

## Author contributions

**Conceptualization:** Richard Muhumuza, Denis Ndekezi, Chiti Bwalya, Musonda Simwinga, Janet Seeley, Andrew Sentoogo Ssemata.

**Data curation:** Richard Muhumuza, Denis Ndekezi, Chiti Bwalya, Joanita Nassali, Jackline Namara, Felix Rutaro, Andrew Sentoogo Ssemata.

**Formal analysis:** Richard Muhumuza, Denis Ndekezi, Rehema Nagawa, Esther Awino, Chiti Bwalya, Madalitso Mbewe, Joanita Nassali, Jackline Namara, Felix Rutaro, Musonda Simwinga, Virginia Bond, Janet Seeley, Andrew Sentoogo Ssemata.

**Funding acquisition:** Andrew Sentoogo Ssemata.

**Investigation:** Richard Muhumuza, Denis Ndekezi, Rehema Nagawa, Esther Awino, Virginia Bond, Janet Seeley, Andrew Sentoogo Ssemata.

**Methodology:** Richard Muhumuza, Rehema Nagawa, Esther Awino, Madalitso Mbewe, Jackline Namara, Felix Rutaro, Musonda Simwinga, Andrew Sentoogo Ssemata.

**Project administration:** Richard Muhumuza, Denis Ndekezi, Chiti Bwalya, Musonda Simwinga, Virginia Bond, Janet Seeley, Andrew Sentoogo Ssemata.

**Resources:** Richard Muhumuza, Andrew Sentoogo Ssemata.

**Software:** Denis Ndekezi.

**Supervision:** Richard Muhumuza, Chiti Bwalya, Madalitso Mbewe, Musonda Simwinga, Virginia Bond, Janet Seeley, Andrew Sentoogo Ssemata.

**Validation:** Richard Muhumuza, Esther Awino, Andrew Sentoogo Ssemata.

**Writing – original draft:** Richard Muhumuza, Andrew Sentoogo Ssemata.

**Writing – review & editing:** Richard Muhumuza, Denis Ndekezi, Rehema Nagawa, Esther Awino, Chiti Bwalya, Madalitso Mbewe, Joanita Nassali, Jackline Namara, Felix Rutaro, Musonda Simwinga, Virginia Bond, Janet Seeley, Andrew Sentoogo Ssemata.

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
