## [Decision Letter · Decision Letter 0]

8 Oct 2025

We look forward to receiving your revised manuscript.

Kind regards,

Gamji Rabiu Abu-Ba'are, Ph.D, MA

Academic Editor

PLOS ONE

Journal Requirements:

https://link.springer.com/article/10.1007/s10461-014-0831-y?

https://researchonline.lshtm.ac.uk/id/eprint/4665167/1/The%20potential%20effect%20of%20pre%20exposure%20prophylaxis%20PrEP%20roll%20out%20on%20sexual%20risk%20behaviour%20among%20adolescents%20and%20young%20people%20in%20East%20and%20southern%20Africa.pdf

In your revision ensure you cite all your sources (including your own works), and quote or rephrase any duplicated text outside the methods section. Further consideration is dependent on these concerns being addressed.

[The study was supported by funding from Wellcome’s Institutional Strategic Support Fund grant 204928/Z/16/Z through the London School of Hygiene and Tropical Medicine. The funders had no role in the study design and decision to publish or preparation of the manuscript.].

4. Thank you for stating the following in your manuscript:

[The study was supported by funding from Wellcome’s Institutional Strategic Support Fund grant 204928/Z/16/Z through the London School of Hygiene and Tropical Medicine. The funders had no role in the study design and decision to publish or preparation of the manuscript.]

[The study was supported by funding from Wellcome’s Institutional Strategic Support Fund grant 204928/Z/16/Z through the London School of Hygiene and Tropical Medicine. The funders had no role in the study design and decision to publish or preparation of the manuscript.]

5. We note that you have indicated that there are restrictions to data sharing for this study. For studies involving human research participant data or other sensitive data, we encourage authors to share de-identified or anonymized data. However, when data cannot be publicly shared for ethical reasons, we allow authors to make their data sets available upon request. For information on unacceptable data access restrictions, please see http://journals.plos.org/plosone/s/data-availability#loc-unacceptable-data-access-restrictions.

6. We note that your paper includes detailed descriptions of individual participants. As per the PLOS ONE policy (http://journals.plos.org/plosone/s/submission-guidelines#loc-human-subjects-research) on papers that include identifying, or potentially identifying, information, the individual(s) or parent(s)/guardian(s) must be informed of the terms of the PLOS open-access (CC-BY) license and provide specific permission for publication of these details under the terms of this license. Please download the Consent Form for Publication in a PLOS Journal (http://journals.plos.org/plosone/s/file?id=8ce6/plos-consent-form-english.pdf). The signed consent form should not be submitted with the manuscript, but should be securely filed in the individual's case notes. Please amend the methods section and ethics statement of the manuscript to explicitly state that the patient/participant has provided consent for publication: “The individual in this manuscript has given written informed consent (as outlined in PLOS consent form) to publish these case details.

Additional Editor Comments:

Thank you for submitting your manuscript to PLOS ONE. The paper addresses an important topic in adolescent HIV prevention and contributes to understanding how youth perceive HIV self-testing in Uganda. Three expert reviewers and the editorial team have carefully evaluated the submission.

After weighing the reviewers’ comments and conducting an independent editorial assessment, I am inviting you to revise and resubmit the manuscript for further consideration. The study demonstrates methodological soundness and relevance, but substantial revisions are required to meet PLOS ONE’s standards for conceptual framing, analytic transparency, and clarity of presentation.

Decision: Major Revision

The study has merit but requires deeper theoretical articulation, clearer reporting of methods and analysis, and stronger synthesis of findings. In addition the responding to the assessment below, please also respond to reviewer comments

Editorial Assessment

While your qualitative design and socio-ecological framing are appropriate, several key issues must be addressed:

1. Conceptual clarity and theoretical depth

• Expand engagement with the Social-Ecological Model (SEM) by mapping each theme to a specific SEM level.

• Briefly justify why SEM is well suited to HIV self-testing concerns, and consider integrating complementary frameworks such as the Health Belief Model or Theory of Planned Behavior to capture perceptions of risk, stigma, and control.

• Avoid implying causality (e.g., that HIVST “causes” risky behavior); retain interpretive focus on perceived or anticipated effects.

2. Methods transparency

• Clearly state the study design (qualitative exploratory).

• Provide the full participant profile (age, gender, schooling, occupation, prior testing, urban/rural site).

• Explain recruitment through VHTs/CHEWs, describing how confidentiality and voluntariness were maintained.

• Describe how data saturation was determined and whether any pilot testing occurred.

3. Analytic procedures

• Re-state the analytic process following Braun & Clarke’s six phases of thematic analysis.

• Clarify how SEM informed coding and theme review without constraining emergent insights.

• Move beyond descriptive quotation to include latent-level interpretation (norms, shame, social pressures).

• Indicate steps taken to ensure credibility (multiple coders, debriefs, audit trail).

4. Reflexivity and positionality

• Retain a brief paragraph summarizing team composition and insider/outsider balance, but focus on how this influenced access, questioning, and interpretation.

5. Contextualization and limitations

• Explain the choice of fishing-community settings and how mobility, economic vulnerability, or social norms shape HIVST use.

• Add a concise Limitations section noting sampling channels that may favor engaged youth, possible social desirability bias, and limited transferability beyond the study sites.

6. Presentation and structure

• Consider merging “Results” and “Discussion” or add a bridging “Findings and Interpretation” section that reconnects themes to the SEM and to existing literature.

• Reduce repetition of quotations; synthesize patterns instead.

• Ensure consistent acronym use (HIVST, VHTs, CHEWs).

• Correct typographical issues (e.g., “goal,” not “gaol”) and apply professional English editing.

7. Ethics and data availability

• Confirm that all anonymized excerpts are included in supplementary materials or a repository, per PLOS ONE data policy.

• Re-affirm IRB approval details and informed-consent procedures.

Reviewers' comments:

Reviewer's Responses to Questions

**Comments to the Author**

1. Is the manuscript technically sound, and do the data support the conclusions?

Reviewer #1: Yes

Reviewer #2: Partly

Reviewer #3: Partly

2. Has the statistical analysis been performed appropriately and rigorously?

Reviewer #1: Yes

Reviewer #2: N/A

Reviewer #3: Yes

3. Have the authors made all data underlying the findings in their manuscript fully available?

Reviewer #1: No

Reviewer #2: Yes

Reviewer #3: No

4. Is the manuscript presented in an intelligible fashion and written in standard English?

Reviewer #1: No

Reviewer #2: Yes

Reviewer #3: Yes

Reviewer #1: Title

Precise and concise, reflects content, target group, and location. Missing study design and study period, consider adding.

Abstract

Includes introduction, objectives, methods, results, and conclusions. Problem and justification not clearly reflected, consider adding. Study design not stated.

Results: include brief participant description (demographics, number of themes, etc.).

Conclusion: highlight major findings first, then recommendations; current text is mostly recommendations.

Background

Explains scientific rationale and context (global and Ugandan HIV/AYP). Knowledge gap clearly stated. Specific aims included.

Line 60: Specify region in Uganda.

Line 61: Check tense, use past tense unless statistics are current.

Line 66: Define HIVST at first mention.

Line 88: Add comma between “testers” and “Despite.”

Aim: clarify focus on experiences, perceptions, and contextual factors influencing HIVST uptake among AYP in Uganda.

Methods

Study design should be explicitly stated (qualitative exploratory study).

Line 130: Define HISTAZU study.

Explain how sample size was determined; comment on saturation.

Clarify participant selection: purposeful or voluntary?

Line 145: Correct spelling of “goal.”

Consistently use “AYP” across the document and not AYPs (plural already).

Data Collection

State language in which interviews were conducted. Specify location and duration of sessions. Was piloting conducted? Were IDIs planned from the start? Explain criteria for selecting 14 IDI participants. Describe coding process: number of coders, cross-checking, consensus, theme development. Clarify whether analysis was inductive or deductive.

Results

Key results summarized along SEC framework, with quotes. Consider adding a visual of key barriers along SEC format.

Line 232: Avoid identifying information in IDI quotes; maintain consistency with GD quotes.

Discussion

Line 461: Change “were” to “where.”

Limitations

Include additional limitations such as bias, small sample size, and self-reporting.

Conclusion

Be precise: focus on main findings, implications, and recommendations. Move extra details to discussion.

Reviewer #2: Article Title: Navigating HIV Self-Testing: concerns among adolescents and young people aged 15-24 years in Uganda

Authors: Richard Muhumuza et al.(2025)

Summary and overall assessment

This qualitative manuscript addresses a high-priority question: how adolescents and young people (15–24) in Uganda navigate HIV self-testing (HIVST). Using a social-ecological framing is appropriate because testing behavior is shaped by factors at individual, interpersonal, community, and system levels. To meet journal standards, the theoretical framing, methods transparency, and analysis need strengthening. Results should connect themes more clearly to theory and to established evidence on youth testing contexts in Uganda. I recommend major revision.

Major strengths

1. Timely topic and population. Youth remain central to epidemic control, and qualitative work can surface barriers and emotions missed by surveys. Recent global statistics underscore the continuing need for youth-friendly testing pathways (UNAIDS, 2025).

2. Level-based structuring. Organizing patterns by social-ecological levels improves clarity and aligns with common practice in HIV prevention research.

3. Attention to underserved groups. A focus on first-time testers and young men is justified; the definition of “marginalized” can be broadened to include LGBTIQ youth and young sex workers who face layered stigma and access barriers in East African settings.

Major issues (required for publication)

1) Theory use and conceptual depth

• Shallow engagement with the social-ecological model (SEM). The SEM is mentioned but not fully used. Map each theme to a specific SEM level and state how the evidence supports, refines, or challenges level-specific propositions. Consider complementary theories to capture latent forces: the Health Belief Model for perceived susceptibility, severity, benefits, barriers, and cues to action (Glanz, Rimer, & Viswanath, 2015); the Theory of Planned Behavior for norms and perceived control (Ajzen, 1991); and stigma/identity perspectives for shame and anticipated discrimination.

• Abductive stance. An abductive approach would allow emergent insights to reshape the conceptual lens rather than forcing data into a fixed frame.

• Chrono-level dynamics. Adolescence and young adulthood involve transitions (school-to-work, marriage, parenthood). Discuss how these transitions alter HIVST meaning over time within a chronosystem view of development.

2) Methods transparency and sample description

• Participant profile. Provide richer demographics to support transferability: age distribution, sex/gender, education, socioeconomic indicators, urban/rural site, prior testing, and (if volunteered) HIV status.

• Recruitment via CHEWs/VHTs. Explain why village meetings and peer mobilization were chosen and how confidentiality and voluntariness were protected when community health cadres supported recruitment. Literature notes role tensions and ethical sensitivities when local health workers engage neighbors on sexual health; show how your procedures mitigated these risks (Lehmann & Sanders, 2007; McCollum et al., 2016).

• Method credibility. Cite precedents using FGDs and peer-led approaches in HIV/STI research with Ugandan youth to anchor these choices.

3) Analysis clarity and alignment

• “Thematic analysis guided by SEM” needs precision. Thematic Analysis (TA) is a flexible method with established steps (familiarization; coding; generating, reviewing, defining/naming themes; write-up). Describe these steps and clarify how SEM functioned as a sensitizing framework rather than a coding template that constrained emergence (Braun & Clarke, 2006).

• From quotations to interpretation. Several sections read as narration of quotes. Strengthen latent-level analysis (norms, shame, social pressure). Define and exemplify each theme and provide concise analytical claims after each excerpt.

• Reflexivity and positionality. The statement about a Zambian–British–Ugandan team is a start, but explain how insider/outsider status, power, culture, and resources shaped access, questioning, and interpretation, and what you did to manage these dynamics (e.g., iterative debriefs, local co-coding, audit trails) (Finlay, 2002).

4) Contextualization of setting and populations

• Fishing communities. Briefly justify relevance: youth in Lake Victoria fishing communities face mobility, economic and social-norm pressures, and higher HIV incidence that can shape HIVST uptake (Kiwanuka et al., 2014).

• Epidemic dynamics. When citing global burden, add one orienting sentence on trends to motivate youth-friendly testing (UNAIDS, 2025).

5) Structure and reporting

• Results vs discussion. In qualitative studies, integrating findings with interpretation often improves coherence. Consider “Findings and Interpretation” or ensure a subsequent Discussion explicitly re-engages theory at each SEM level.

• Limitations. Add a clear limitations section (sampling channels may favour more engaged youth; social desirability in FGDs; transferability beyond sampled districts).

• Editing. Correct typos (e.g., “goal,” not “gaol”) and tighten long sentences.

Minor issues (actionable edits)

• Define “marginalized” to include LGBTIQ youth and young sex workers (where lawful and safe), with ethical safeguards noted.

• Spell out acronyms at first use (HIVST, CHEWs, VHTs).

• Tag participant quotes (FGD number, participant code) for transparency.

Specific, testable revision checklist

1. Revise Conceptual Framework to specify SEM assumptions by level; add 1–2 complementary theories (HBM, TPB); justify an abductive stance.

2. Expand Methods with full participant profile, recruitment rationale, ethical safeguards with CHEWs/VHTs, and citations to similar Ugandan youth HIVST/peer-led designs.

3. Re-state Analysis following Braun and Clarke’s six phases; explain how SEM informed memos/theme review without constraining emergence; note audit-trail elements.

4. Re-structure Results/Discussion so each theme is defined, evidenced with quotes, interpreted at manifest and latent levels, linked back to SEM and complementary theories, and situated against prior Ugandan youth evidence.

5. Add a Context paragraph on fishing communities to justify salience.

6. Insert a Limitations section on sampling, desirability bias, and transferability.

7. Update Background with current burden and trajectory to motivate youth-friendly testing.

References

Ajzen, I. (1991). The theory of planned behavior. Organizational Behavior and Human Decision Processes, 50(2), 179–211.

Braun, V., & Clarke, V. (2006). Using thematic analysis in psychology. Qualitative Research in Psychology, 3(2), 77–101.

Finlay, L. (2002). Negotiating the swamp: The opportunity and challenge of reflexivity in research practice. Qualitative Research, 2(2), 209–230.

Glanz, K., Rimer, B. K., & Viswanath, K. (Eds.). (2015). Health behavior: Theory, research, and practice (5th ed.). San Francisco, CA: Jossey-Bass.

Kiwanuka, N., Ssetaala, A., Nalutaaya, A., Mpendo, J., Wambuzi, M., Nanvubya, A., ... & Kaleebu, P. (2014). High HIV-1 prevalence, risk behaviours, and willingness to participate in HIV prevention trials in fishing communities on Lake Victoria, Uganda. PLoS ONE, 9(5), e94932.

Lehmann, U., & Sanders, D. (2007). Community health workers: What do we know about them? Geneva: World Health Organization.

McCollum, R., Gomez, W., Theobald, S., & Taegtmeyer, M. (2016). How equitable are community health worker programmes and which programme features influence equity of community health worker services? A systematic review. BMC Public Health, 16, 419.

UNAIDS. (2025). Global AIDS Update 2025. Geneva: Joint United Nations Programme on HIV/AIDS.

Reviewer #3: As a researcher I would give a 60% to the author ,more sustainable measures need to be arisen especially on the area of children and also more realistic regressions and Correlations need to be assessed.

Deploy more econometric measures but other wise all is well

**Do you want your identity to be public for this peer review?** For information about this choice, including consent withdrawal, please see our Privacy Policy

Reviewer #1: No

Reviewer #2: **Yes:** Musisi John Kaduwanema

Reviewer #3: **Yes:** Uwayesu Happy E

---

## [Author Response · Author response to Decision Letter 1]

9 Dec 2025

20th November 2025.

To: The Editor,

PLOS One

Dear Editor,

RE: RESPONSE TO EDITOR AND REVIEWERS’ COMMENTS

We are pleased to resubmit our manuscript entitled “Navigating HIV Self-Testing: Concerns among Adolescents and Young People aged 15–24 years in Uganda: An Exploratory Qualitative Study” for further consideration in PLOS ONE.

We sincerely thank the editors and reviewers for their careful evaluation and constructive feedback. We have revised the manuscript extensively in response to the comments provided and believe these revisions have substantially strengthened the clarity, rigor, and contribution of the work. Below, we provide a detailed, point-by-point response to all reviewer comments. For each comment, we indicate how and where the corresponding changes have been addressed in the revised manuscript.

Additionally, I confirm that all relevant data are contained within the manuscript and Supporting Information files. All raw data used in the analyses have been provided to enable replication of the study in accordance with PLOS ONE data availability requirements

Reviewer 1:

While your qualitative design and socio-ecological framing are appropriate, several key issues must be addressed:

1. Conceptual clarity and theoretical depth

• Expand engagement with the Social-Ecological Model (SEM) by mapping each theme to a specific SEM level.

Response: Thank you for this comment, we have revised the model to reflect the level at which all themes are – See table 2 in the results section. Line Number 326.

• Briefly justify why SEM is well suited to HIV self-testing concerns, and consider integrating complementary frameworks such as the Health Belief Model or Theory of Planned Behavior to capture perceptions of risk, stigma, and control.

Response: Thank you for this thoughtful suggestion. We have now clarified why the Social Ecological Model (SEM) is well suited to examining HIV self-testing concerns among adolescents and young people, and we have integrated complementary insights from the Health Belief Model (HBM) and Theory of Planned Behaviour (TPB) to strengthen the theoretical grounding. We have incorporated this explanation into the revised manuscript (Methods section, under “Theoretical Framework”) to demonstrate how SEM provides the overarching structure, while HBM and TPB offer complementary psychological insights, thereby strengthening the interpretation of adolescents’ concerns around HIV self-testing in Wakiso District. Line number 118-151

• Avoid implying causality (e.g., that HIVST “causes” risky behavior); retain interpretive focus on perceived or anticipated effects.

Response: Thank you for this important observation. We agree that our qualitative data reflects perceived or anticipated effects of HIV self-testing rather than causal relationships. In response, we have revised the relevant sections of the manuscript to remove causal language and to ensure that we frame participants’ concerns as perceptions, fears, or anticipated outcomes.

2. Methods transparency

• Clearly state the study design (qualitative exploratory).

Response: Thank you for this comment, we have stated the study design as an exploratory qualitative study. Refer to line number 155

• Provide the full participant profile (age, gender, schooling, occupation, prior testing, urban/rural site).

Response: Thank you for the comment. We provided the full profile of the participants in the demographic characteristics of the participants. Refer to table 1 in the manuscript.

• Explain recruitment through VHTs/CHEWs, describing how confidentiality and voluntariness were maintained.

Response: Thank you for the comment. We have revised the recruitment section to clearly explain the specific role of VHTs/CHEWs and how confidentiality and voluntariness were ensured. As now described in the manuscript, CHEWs supported only with initial community mobilisation and were not present during information sessions, consent processes, or any study procedures. All recruitment and enrolment were conducted privately by the research team to minimise influence and protect participant confidentiality. See participant selection sectionRefer to Line number 187-202.

• Describe how data saturation was determined and whether any pilot testing occurred.

Response: Additional text has been added to main document describing how data saturation was determined both at the level of data collection and analysis. Refer to Line number 277 and Line number 250-251 respectively.

3. Analytic procedures

• Re-state the analytic process following Braun & Clarke’s six phases of thematic analysis.

Response: Thank you for the comment. We followed Braun and Clarke’s six phases of thematic analysis to guide the analytic process. Throughout the analysis, the Social Ecological Model (SEM) served as a sensitising framework that enabled the team to consider influences at individual, interpersonal, community, and structural levels. SEM did not determine the coding structure but supported an integrated inductive–deductive approach that allowed themes to emerge from participants’ narratives while also guiding their interpretation. A text has been added to explain the steps used in analysis of the data Line Number 235-245

• Clarify how SEM informed coding and theme review without constraining emergent insights.

Thank you for the comment. The SEM was used as a sensitising framework during analysis. It did not determine or restrict the coding structure; instead, it helped the team remain attentive to influences operating at individual, interpersonal, community, and structural levels as themes emerged inductively from the data. Line number 250-255

• Move beyond descriptive quotation to include latent-level interpretation (norms, shame, social pressures).

Response: We thank you for this suggestion. We have revised the Results section to integrate latent-level interpretation alongside participants’ quotations. We now discuss how aspects such as social norms, stigma, familial authority, and structural inequities shape adolescents’ and young people’s perceptions of HIV self-testing.

• Indicate steps taken to ensure credibility (multiple coders, debriefs, audit trail).

Response: Thank you for this helpful suggestion. We have revised the Methods section to more clearly outline the steps taken to ensure credibility and rigor in the qualitative analysis. The revised text appears in the Data Analysis subsection. Line number 246-250

4. Reflexivity and positionality

• Retain a brief paragraph summarizing team composition and insider/outsider balance, but focus on how this influenced access, questioning, and interpretation.

Response: We have revised the reflexivity and positionality section to clarify how team composition influenced the study. The Ugandan team, as insiders, facilitated access to adolescents and young people and conducted culturally sensitive interviews, while the Zambian and British team members provided outsider perspectives that enhanced reflexive interpretation during analysis.

5. Contextualization and limitations

• Explain the choice of fishing-community settings and how mobility, economic vulnerability, or social norms shape HIVST use.

Response: We have added text in the data that shows that AYP in the fishing communities are at high risk of contracting HIV and could therefore benefit from HIVST in the methods section.

• Add a concise Limitations section noting sampling channels that may favor engaged youth, possible social desirability bias, and limited transferability beyond the study sites.

Response: This has been added to the limitation section of the manuscript.

6. Presentation and structure

• Reduce repetition of quotations; synthesize patterns instead.

Response: we have reduced the quotes and synthesised the narratives throughout the results section.

• Ensure consistent acronym use (HIVST, VHTs, CHEWs).

Response: Thank you and these have been harmonised.

• Correct typographical issues (e.g., “goal,” not “gaol”) and apply professional English editing.

Response: Thank and this has been harmonised as well.

7. Ethics and data availability

• Confirm that all anonymized excerpts are included in supplementary materials or a repository, per PLOS ONE data policy.

I confirm that all anonymized excerpts are included in the results section and all relevant data are contained within the manuscript.

Reviewer #2: Title

Precise and concise, reflects content, target group, and location. Missing study design and study period, consider adding.

Response: Study design has been added to the title.

Abstract

Includes introduction, objectives, methods, results, and conclusions. Problem and justification not clearly reflected, consider adding. Study design not stated.

Results: include brief participant description (demographics, number of themes, etc.).

Conclusion: highlight major findings first, then recommendations; current text is mostly recommendations.

Response: Thank you for the observation. We have harmonized the abstract.

Background

Line 61: Check tense, use past tense unless statistics are current.

Response: Thank you for this observation, we have changed it to past tense.

Line 66: Define HIVST at first mention.

Response: We had earlier defined it in the abstract but nevertheless, we have written it in full in this section.

Line 88: Add comma between “testers” and “Despite.”

Response: Thank you for the observation, we have broken down the sentence to read better. See Line 95-98.

Methods

Study design should be explicitly stated (qualitative exploratory study).

Response: Thank you and this has been stated.

Line 130: Define HISTAZU study.

Response: Thank you and this has been explained in the methods section. Line number 193-198

Clarify participant selection: purposeful or voluntary?

Response: We have stated that we used purposive sampling to recruit our participants. See line 200-203

Line 145: Correct spelling of “goal.”

Response: Thank you for the careful review, the spelling error has been corrected. See line 278

Consistently use “AYP” across the document and not AYPs (plural already).

Response: This has been harmonised. Thanks!

Data Collection

State language in which interviews were conducted. .Specify location and duration of sessions. Was piloting conducted? Were IDIs planned from the start? Explain criteria for selecting 14 IDI participants. Describe coding process: number of coders, cross-checking, consensus, theme development. Clarify whether analysis was inductive or deductive

Response: This has been reworked in the methodology section see line 168-271 and 311-330

Results

Key results summarized along SEC framework, with quotes. Consider adding a visual of key barriers along SEC format.

Response: We have added Table 2 to summarise the results. See line 458

Line 232: Avoid identifying information in IDI quotes; maintain consistency with GD quotes.

Response: All quotes have been anonymised and are now consistent.

Discussion

Line 461: Change “were” to “where.”

Response: Thank you and this has been changed.

Limitations

Include additional limitations such as bias, small sample size, and self-reporting.

Response: We have revised the strength and limitations section. See line 822-836

Conclusion

Be precise: focus on main findings, implications, and recommendations. Move extra details to discussion.

Response: We have revised the conclusion section. See line 838-850

Major issues (required for publication)

1) Theory use and conceptual depth

• Shallow engagement with the social-ecological model (SEM). The SEM is mentioned but not fully used. Map each theme to a specific SEM level and state how the evidence supports, refines, or challenges level-specific propositions. Consider complementary theories to capture latent forces: the Health Belief Model for perceived susceptibility, severity, benefits, barriers, and cues to action (Glanz, Rimer, & Viswanath, 2015); the Theory of Planned Behavior for norms and perceived control (Ajzen, 1991); and stigma/identity perspectives for shame and anticipated discrimination.

Response: A more detailed text has been added to the manuscript to explicitly map each theme and also incorporated the HBM and TPB.

• Abductive stance. An abductive approach would allow emergent insights to reshape the conceptual lens rather than forcing data into a fixed frame.

Response: We appreciate the reviewer’s suggestion regarding an abductive analytic stance. We have revised the Methods section to clarify that while SEM provided a sensitizing framework, we adopted an abductive approach, allowing emergent insights from participants’ narratives to shape and refine our conceptual lens.

• Chrono-level dynamics. Adolescence and young adulthood involve transitions (school-to-work, marriage, parenthood). Discuss how these transitions alter HIVST meaning over time within a chronosystem view of development.

Response: We thank the reviewer for highlighting the importance of chrono-level dynamics. We have added discussion of how adolescents’ and young adults’ developmental transitions such as school-to-work transitions, partnership formation, and entry into parenthood affect the perceived meaning, risks, and benefits of HIV self-testing in the discussion section of the manuscript. This situates our findings within the SEM’s chronosystem and underscores the need for age-appropriate support strategies Line number 569-581

2) Methods transparency and sample description

• Recruitment via CHEWs/VHTs. Explain why village meetings and peer mobilization were chosen and how confidentiality and voluntariness were protected when community health cadres supported recruitment. Literature notes role tensions and ethical sensitivities when local health workers engage neighbors on sexual health; show how your procedures mitigated these risks (Lehmann & Sanders, 2007; McCollum et al., 2016).

Response: We appreciate the reviewer’s comment regarding recruitment through CHEWs/VHTs. We have clarified in the manuscript why village meetings and peer mobilization were chosen and how confidentiality and voluntariness were maintained. Specifically, CHEWs only supported by informing the community about upcoming meetings; they were not present during meetings or involved in any study procedures, minimizing participation bias and protecting privacy. Interested adolescents and young people voluntarily approached the research team for enrolment after receiving full information about the study. Line number 187-202

• Method credibility. Cite precedents using FGDs and peer-led approaches in HIV/STI research with Ugandan youth to anchor these choices.

Response: This has been explained in the methods section.

3) Analysis clarity and alignment

• “Thematic analysis guided by SEM” needs precision. Thematic Analysis (TA) is a flexible method with established steps (familiarization; coding; generating, reviewing, defining/naming themes; write-up). Describe these steps and clarify how SEM functioned as a sensitizing framework rather than a coding template that constrained emergence (Braun & Clarke, 2006).

Response: This has been explained in the main document. Line Number 251-255

• From quotations to interpretation. Several sections read as narration of quotes. Strengthen latent-level analysis (norms, shame, social pressure). Define and exemplify each theme and provide concise analytical claims after each excerpt.

Response: Thank you for the great suggestion. We have reworked the results section to reflect these.

• Reflexivity and positionality. The statement about a Zambian–British–Ugandan team is a start, but explain how insider/outsider status, power, culture, and resources shaped access, questioning, and interpretation, and what you did to manage these dynamics (e.g., iterative debriefs, local co-coding, audit trails) (Finlay, 2002).

Response: We have reworked the positionality and reflexive section. Line Number 276-287

4) Contextualization of setting and populations

• Fishing communities. Briefly justify relevance: youth in Lake Victoria fishing communities face mobility, economic and social-norm pressures, and higher HIV incidence that can shape HIVST uptake (Kiwanuka et al., 2014).

---

## [Decision Letter · Decision Letter 1]

22 Jan 2026

Navigating HIV Self-Testing: concerns among adolescents and young people aged 15-24 years in Uganda. An exploratory qualitative study.

PONE-D-25-40176R1

Dear Author,

We’re pleased to inform you that your manuscript has been judged scientifically suitable for publication and will be formally accepted for publication once it meets all outstanding technical requirements.

Kind regards,

Gamji Rabiu Abu-Ba'are, Ph.D, MA

Academic Editor

PLOS One

Additional Editor Comments (optional):

Congratulations, all reviews have been addressed and I recommend your manuscript for publication with PLOS ONE

Reviewers' comments:

Reviewer's Responses to Questions

**Comments to the Author**

Reviewer #1: All comments have been addressed

Reviewer #2: All comments have been addressed

2. Is the manuscript technically sound, and do the data support the conclusions?

Reviewer #1: Yes

Reviewer #2: Yes

3. Has the statistical analysis been performed appropriately and rigorously?

Reviewer #1: Yes

Reviewer #2: N/A

4. Have the authors made all data underlying the findings in their manuscript fully available?

Reviewer #1: Yes

Reviewer #2: Yes

5. Is the manuscript presented in an intelligible fashion and written in standard English?

Reviewer #1: Yes

Reviewer #2: Yes

Reviewer #1: Title and abstract: Title and the abstract modified as guided.

Background: It has been re-writted, the aim of the study is clearer and the typos were removed.

Methods: All identified gaps were addressed.

Results and discussion: All comments addressed.

Reviewer #2: Dear authors. I am absolutely delighted with the changes you have so thoughtfully made to your work. This is very important research as we all participate in creating a society without HIV. Thank you and wishing you all the best in your publication journeys. Warmest regards. Musisi John.

**Do you want your identity to be public for this peer review?** For information about this choice, including consent withdrawal, please see our Privacy Policy

Reviewer #1: **Yes:** Kayinda Francis

Reviewer #2: **Yes:** Musisi John Kaduwanema

---

## [Editor Report · Acceptance letter]

PONE-D-25-40176R1

PLOS One

Dear Dr. Muhumuza,

I'm pleased to inform you that your manuscript has been deemed suitable for publication in PLOS One. Congratulations! Your manuscript is now being handed over to our production team.

Kind regards,

on behalf of

Dr. Gamji Rabiu Abu-Ba'are

Academic Editor

PLOS One